# Others’ Facial Expressions Influence Individuals Making Choices and Processing Feedback: The Event-Related Potential and Behavioral Evidence

**DOI:** 10.3390/ijerph20010568

**Published:** 2022-12-29

**Authors:** Xin Yu, Bo Xu, Entao Zhang

**Affiliations:** 1Institute of Cognition, Brain & Health, Henan University, Kaifeng 475001, China; 2Institute of Psychology and Behavior, Henan University, Kaifeng 475001, China

**Keywords:** trust game, facial expressions, making choices, feedback, ERP

## Abstract

To date, several studies have found the effect of facial expressions on trust decision, using the event-related potentials (ERPs). However, little is known about the neural mechanism underlying the modulation effect of facial expressions on making choices and subsequent outcome evaluation. In the present study, using an ERP technique, we investigated how the neural process of making choices and subsequent outcome evaluation were influenced by others’ facial expressions for the first time. Specifically, participants played a modified version of the Trust Game, in which they watched a photo of the trustee before making choices. Critically, trustees’ faces differed regarding emotional types (i.e., happy, neutral, or angry) and gender (i.e., female or male). Behaviorally, an interaction between expressions and gender was observed on investment rates. On the neural level, the N2 and P3 amplitudes were modulated by facial expressions in the making-choice stage. Additionally, the feedback-related P3 was also modulated by facial expressions. The present study proved the effect of facial expressions on making choices and subsequent outcome evaluation.

## 1. Introduction

Trust is the willingness of a party comprising the intention to accept vulnerability based upon positive expectations of the intentions or the behavior of another [1]. Trust is a prerequisite for successful social relations. In social interactions, the faces of strangers represent a complex visual pattern, which conveys the information of race, expression, gender, and age [2]. Some studies have shown that these cues can trigger stereotypes and influence the formation of beliefs, which can be used to generate initial trust and plan follow-up actions [3]. In addition, humans have developed the ability to extract information from others’ faces to forecast their follow-up actions in a directly or indirectly way [4]. In brief, people often use others’ facial expressions to predict their likely behavior in social interactions [5,6,7,8,9].

Researchers, using the Trust Game (TG) paradigm [10], found the influence of others’ facial expressions on individual decision-making. This game is a very suitable behavioral paradigm for studying decision making, with at least two players (a trustor and a trustee). At the beginning of the game, players receive a sum of initial endowment. Next, the trustor has two options: one is to invest the money on the trustee, and then the initial endowment is multiplied by an amount; another is to keep, and then the game is over. Finally, if the trustor chooses to invest, it is the trustee’s turn to make a choice. They also have two choices: one is to share half of the income (the received money plus the initial endowment) with the trustor, and another is to swallow all the money. A previous study, especially, using the TG paradigm, found that smiling trustees were more trusted than non-smiling trustees [5]. Similarly, other previous studies reported that angry trustees elicited lower trust rates compared with happy and neutral trustees [6,7]. Furthermore, a study found that trust decisions were still affected by expressions of trustees, even if participants were asked to ignore facial expressions [8]. Recently, Ewing and his colleagues [9] found that children were more trusting of happy trustees compared with neutral and angry trustees, as early as five years of age.

In addition to the behavioral experiments, several researchers, using the event-related potential (ERP) techniques, further examined the neural mechanisms of the effect of facial expressions on decision making. For example, an ERP study [11] found that the modulations of expressions on N2 and N3 were observed before the decision-making stage, suggesting that the brain processed emotional expressions before making choices in the Trust Game. Tortosa and his colleagues found the effect of expressions on the face process stage and the decision-making stage, as indexed by N170, P2, and P3 [12]. Additionally, Sonia et al. found that the expressions modulated the central N1 and VPP potentials on the face-perception stage and the influence of expressions on P3b was observed during the decision-making stage [13]. In fact, Gu and his colleagues [14] argued that the process of decision-making can be divided into evaluating available options (co-operator assessment), making choices (behavioral output), and then receiving outcome (feedback processing). However, according to the standard of division [14], the above three ERP studies mainly focus on the available option and making-choices stage, but ignore the outcome-evaluation stage during trust decision making. Therefore, in the present study, we investigated the influences of facial expressions on both the making-choices and feedback stages by using ERP techniques. Additionally, gender was also investigated in the present study, as it was also an important facial clue. Compared with the latest recent research about facial expressions [15], the present research used the TG paradigm, which created a more realistic decision-making environment. Therefore, the result would be beneficial for guiding individuals to make rational decisions in real social interactions. Meanwhile, the study [15] focused on the gender of the participants. By contrast, we extended the study [15] by exploring the effect of the partner’s gender on decision making.

During the making-choices stage, previous studies mainly investigated N2 and P3 components [16]—specifically, the N2, which is a negative ERP deflection over frontocentral brain regions with a 250–350 ms peak latency. Some studies found that more negative N2 amplitudes were induced by high-conflict situations compared with low-conflict situations [17,18]. Therefore, the N2 is considered to be an index of conflict monitoring [19]. Following N2 is the P3, which is a positive component with a 300–500 ms peak latency at central and parietal brain regions. The P3 is considered to be an index of prosocial motivation, as more positive P3 amplitudes were found to be associated with high prosocial behavior than low prosocial behavior [20]. During the feedback stage, previous studies mainly investigated FRN and P3 components [21]. Specifically, the FRN is a negative component with a 250–350 ms peak latency at frontal brain regions. Experimental evidence suggested that more negative FRN amplitudes were induced by loss than gain, and then the FRN is considered to be an index of negative and unexpected outcomes [22,23]. Following FRN, the feedback-related P3 is a positive component with a 300–500 ms peak latency. The functional significance of the P3 component is still controversial. Some researchers found that more positive P3 amplitudes were induced by gain than loss feedback [24,25]. However, other researchers found that the P3 amplitudes were related to the absolute magnitude of the feedback outcome, whether it concerns a gain or a loss of money [26,27].

In order to investigate the influence of others’ expressions on decision making, participants played a modified version of the Trust Game, in which they watched a photo of the trustee before making choices. At the same time, the behavioral and electrophysiological data of participants were recorded. In sum, according to the above review of ERP studies, we expected facial expressions and gender to have an effect on the making-choices (N2 and P3) and feedback stage (FRN and P3), and affect the cooperation rates of the observers, at least for facial expressions. The results of this study may provide evidence for exploring the neural mechanism of making choices and processing feedback. Meanwhile, the result would be beneficial for guiding individuals to make rational decisions in real social interactions.

## 2. Methods

### 2.1. Participants

In order to estimate the sample size, we used G*Power 3.1 software to conduct a priori power analysis for a 2 × 3 within-participants repeated-measures analysis of variance (ANOVA, SPSS 25.0 (IBM Corp., Armonk, NY, USA)). The results showed that at least 19 participants were required to achieve a power of 0.80. In addition, the other parameters included an expected effect size of at least 0.25 (f), an alpha of 0.05, a default within-subject measurement correlation of 0.5, and a non-sphericity correlation value (ε) of 1. In fact, 25 participants, who had normal or corrected-to-normal vision and were all right-handed, were recruited at the Henan University. All participants reported no history of affective disorders and were free of any psychiatric medication. Finally, the EEG data of 23 participants (12 females; mean age 20.17) were used in the following analysis, because the other two participants had more than 50% of the trials being eliminated due to artifacts and excessive noise. This study was conducted in accordance with the guidelines of the Declaration of Helsinki. Meanwhile, the studies involving human participants were reviewed and approved by the Research Ethics Committee of the Institute of Psychology and Behavior, Henan University (protocol code 20210910001 at 10 September 2021). All participants provided written informed consent prior to the experiment.

### 2.2. Materials

According to different facial expressions and gender, face pictures were divided into six types in the present study. Six pictures were selected as experimental materials for each type, with a total of 36, taken from the Chinese Facial Affective Picture System [28]. We invited 20 students (10 females, mean age 20.8), who had not participated in the formal experiment, to evaluate the pictures through the following two questions: “1. What expression does the face picture show?” “2. Please use a 9-point scale, from 1 (no feeling) to 9 (very strong), to evaluate the intensity of the expression in the picture.”

The first question is to investigate whether different types of expressions can cause corresponding emotional perception. According to the evaluation results of 20 students, the emotional accuracy was above 80% for angry, happy, and neutral faces. The second question estimated the emotional arousal of six types of pictures, which found the main effect of expressions, *F* (2, 38) = 18.311, *p* < 0.001, $\eta_{p}^{2}$ = 0.969, suggesting that happy faces (4.825 ± 0.077; *p* < 0.001) and angry faces (4.887 ± 0.180; *p* < 0.001) were rated higher emotional arousal than neutral faces (2.613 ± 0.054), while there were no differences between happy and angry faces (*p* = 1.000). In addition, there were no differences between female (4.131 ± 0.094) and male faces (4.086 ± 0.103) on emotional arousal, *F* (1, 19) = 0.157, *p* = 0.709. Additionally, the interaction between expressions and gender was not significant, *F* (2, 44) = 2.698, *p* = 0.078. These results indicated that expressions and gender of pictures elicited a medium level of emotions. These 36 pictures (8 cm × 11 cm pixel width and height, respectively) were used as faces of trustees for the ERP experiment (see Figure 1).

### 2.3. Task and Procedure

Prior to the experiment, participants learned that they needed to decide if they would co-operate with trustees, represented by different emotional types and gender (see Figure 1). On each trial, the game players received 10 game points at first. Next, the trustor has two options: one is to invest all 10 points in the trustee, and then the 10 points are multiplied by 3; another is to keep the points, and then the game is over. Finally, if the trustor chooses to invest, it is the trustee’s turn to make the choice. They also have two choices: one is to share half of the income (40 game points) with the trustee, and another is to swallow all the money. We told the participants that the answers of the trustees, in each round, were randomly selected from the previously collected databases to convince the participants that they were interacting with real people. However, in fact, the trustee’s answers were based on a computer program, which had a 50% chance of choosing to distribute the money equally.

In the experiment, participants were seated in a quiet room approximately 100 cm from a computer screen and completed 360 rounds of the Trust Game, while their brain potentials were recorded using electroencephalograms (EEGs). To familiarize participants with the task, the experiment started with 12 practice trials. Each trial was initiated by a small white cross presented for a 400–600 ms duration on a black screen (see Figure 1); then, a face appeared in the center of the screen for 1000 ms. After a variable 400 to 600 ms fixation cross, a picture displayed decision options in the center of the screen for 2000 ms; participants chose to either keep (cued by the number 1) or invest (cued by the number 3) the initial endowment by using their index finger to press either the 1 or 3 key on the keyboard. The key positions (1 or 3) of keep (distrust) and invest (trust) were counterbalanced between participants. If participants failed to respond within 2000 ms, a new trial was provided for them to input a valid response. Following a variable 800 to 1000 ms interval with a black screen, the outcome feedback (0 or 20) of participants’ current trial was displayed for 1000 ms. The gain/loss outcomes were determined in pseudorandom sequence, with half of them gain trials and the other half loss trials. However, participants were not told about these manipulations. Each facial picture was randomly repeated ten times, and every face was equally associated with a win or loss. The task consisted of six blocks with 60 trials each. E-prime 3.0 was used for stimuli presentation, sending markers, and response recording.

### 2.4. Behavioral Recording and Analysis

We used a repeated-measures ANOVA to examine the difference between expressions and gender in participants’ investment rates and reaction time (RT). A two-way repeated measure analysis of variance (ANOVA) was conducted on investment rates by SPSS software (IBM, Armonk, NY, USA), with expressions (happy, neutral, angry) and gender (female vs. male) as within-subject factors. In addition, a three-way ANOVA was conducted on RT, with the within-subject factors of expressions (happy, neutral, angry), gender (female, male), and trust choices (trust, distrust).

### 2.5. Electrophysiological Recording and Analysis

We used the 64 Ag/AgCl electrode cap (Brain Product), whose electrodes were arranged according to the standard 10–20 system, with a sampling rate of 500 Hz, to collect- the electroencephalography (EEG) data. Meanwhile, the left and right mastoids were taken as references. Vertical electrooculogram (EOG) was recorded supraorbitally and infraorbitally from the right eye. The impedance of all electrodes was kept less than 10 kΩ. The horizontal EOG was recorded as left versus right orbital rim. The EEG and EOG measurements were amplified using a 0.1–30 Hz bandpass and continuously digitized at 500 Hz for offline analysis.

The Brain Vision Analyzer (Brain Products, Germany) was used for offline analysis. Independent Component Analysis (ICA) removed ocular artifacts. All epochs in which EEG voltages exceeded a threshold of ±80 μV were excluded from further processing. During the making-choices stage, about 30 effective trials remained for each condition in each participant. During the feedback stage, gender was excluded from all analyses due to insufficient effective trials. Finally, about 30 effective trials remained for each condition in each participant. The EEG was time-locked to the onset of decision choices and feedback stimuli, and from 200 ms pre-stimulus to 800 ms post-stimulus.

Based on the existing literature [29], the following 15 electrode sites (frontal: Fz, F3, F4; fronto-central: FCz, FC3, FC4; central: Cz, C3, C4; centro-parietal: CPz, CP3, CP4; and parietal: Pz, P3, and P4) were selected to calculate, respectively, maximum electrode site of each ERP component. Then, the follow-up statistical analysis was calculated on the electrode with maximum. According to the result of calculation, the N2 amplitude was larger at the Fz site (−2.29 µV) than other sites. Therefore, the N2 was measured as mean amplitudes in a 100 ms time window around its peak (250–350 ms) at electrode Fz [30]. Following the same methods, during the making-choices stage, the P3 was measured as mean amplitudes between 250 and 400 ms at electrode P3, Pz, and P4 [31], whose voltage was subsequently averaged. In this stage, a three-way repeated measure analysis of variance (ANOVA) was conducted on each component, with the within-subject factors of expressions (happy, neutral, angry), gender (female, male), and trust choices (trust, distrust). During the feedback stage, the FRN was measured as mean amplitudes between 270 and 310 ms at Fz [32], and the P3 was measured as mean amplitudes between 300 and 400 ms at Pz [29]. In this stage, a two-way repeated measure analysis of variance (ANOVA) was conducted on each component, with the within-subject factors of expressions (happy, neutral, angry) and feedback (gain, loss).

Statistical analysis was performed using SPSS 25.0. The *p* values of all the main effects and interactions were corrected by applying the Greenhouse–Geisser method when needed. Post-hoc testing of significant main effects was conducted with the Bonferroni correction method. Significant interactions were further examined using simple-effect analysis, and the partial eta-squared, which was a measure of the proportion of variance, was also reported. The proportion of variance was explained by the independent variable.

## 3. Results

### 3.1. Behavioral Data

A 3(expressions: happy, neutral, angry) × 2(gender: female, male) repeated-measures ANOVA were conducted on investment rates and showed a significant main effect of expressions, *F* (2, 44) = 18.311, *p* < 0.001, $\eta_{p}^{2}$ = 0.454, suggesting that happy faces (0.651 ± 0.034) elicited the highest investment rate, followed by neutral faces (0.552 ± 0.037; *p* = 0.011), then angry faces (0.347 ± 0.045). The interaction between expressions and gender reached marginally significant, *F* (2, 44) = 2.698, *p* = 0.078, $\eta_{p}^{2}$ = 0.454. The simple effect analyses demonstrated that happy (0.628 ± 0.038; *p* = 0.001) and neutral faces (0.558 ± 0.037; *p* = 0.002) elicited higher investment rates than angry faces (0.352 ± 0.046) when participants saw female trustees, while the difference between happy and neutral faces was absent (*p* = 0.121). In contrast, happy faces (0.673 ± 0.033) elicited the highest investment rate, followed by neutral faces (0.545 ± 0.041; *p* = 0.003), then angry faces (0.341 ± 0.048; *p* < 0.001), when participants saw male trustees. A main effect of gender was not significant *F* (1, 22) = 0.136, *p* = 0.716 (see Table 1).

A 3(expressions: happy, neutral, angry) × 2(gender: female, male) × 2(decision choices: trust, distrust) analysis of variance (ANOVA) was conducted on the reaction times (RT). We did not find any main effects or interaction effects (*p* > 0.05; see Table 2).

### 3.2. ERP Data

#### 3.2.1. Making-Choices Stage

N2. A 3(expressions: happy, neutral, angry) × 2(gender: female, male) × 2(decision choices: trust, distrust) repeated-measures ANOVA of the N2 average amplitudes showed a marginal significant interaction between expressions and gender, *F* (2, 44) = 2.880, *p* = 0.067, $\eta_{p}^{2}$ = 0.116. The simple effect analyses demonstrated that neutral faces (−2.216 ± 0.359 µV) elicited a larger N2 amplitude than angry faces (−1.378 ± 0.436 µV, *p* = 0.028) when participants saw female trustees, while happy faces (−1.746 ± 0.482 µV) did not differ from angry or neutral faces (*p* > 0.05). In contrast, the amplitude differences among angry (−2.052 ± 0.492 µV), happy (−1.676 ± 0.567 µV), and neutral faces (−1.912 ± 0.532 µV) were not significant when participants saw male trustees (all *p* > 0.05).

However, the main effect of expressions *F* (2, 44) = 1.151, *p* = 0.326, gender *F* (1, 22) = 0.274, *p* = 0.606, and decision choices *F* (1, 22) = 1.094, *p* = 0.307 were not significant. In addition, the interaction between expressions and decision choices, *F* (2, 44) = 1.008, *p* = 0.373, the interaction between gender and decision choices, *F* (1, 22) = 0.239, *p* = 0.629, and the three-way interaction among expressions, gender, and decision choices were not significant, *F* (2, 44) = 0.844, *p* = 0.437.

P3. A 3(expressions: happy, neutral, angry) × 2(gender: female, male) × 2(decision choices: trust, distrust) repeated-measures ANOVA of the P3 average amplitudes showed a marginally significant interaction between expressions and decision choices, *F* (2, 44) = 2.895, *p* = 0.66, $\eta_{p}^{2}$ = 0.116. The simple effect analyses demonstrated that happy faces (2.390 ± 0.337 µV) elicited a larger P3 amplitude than neutral faces (1.814 ± 0.280 µV, *p* = 0.070) for distrust choices, while angry faces (2.103 ± 0.311 µV) did not differ from happy or neutral faces (*p* > 0.05). In contrast, the amplitude differences among angry (2.122 ± 0.326 µV), happy (2.046 ± 0.315 µV), and neutral faces (2.289 ± 0.311 µV) were not significant for trust choices (all *p* > 0.05).

However, the main effect of expressions *F* (2, 44) = 0.676, *p* = 0.514, gender *F* (1, 22) = 0.002, *p* = 0.965, and decision choices *F* (1, 22) = 0.061, *p* = 0.807 were not significant. In addition, the interaction between expressions and gender, *F* (2, 44) = 1.308, *p* = 0.281, the interaction between gender and decision choices, *F* (1, 22) = 1.676, *p* = 0.209, and the three-way interaction among expressions, gender, and decision choices were not significant, *F* (2, 44) = 1.370, *p* = 0.265 (see Figure 2).

#### 3.2.2. Feedback Stage

FRN. A 3(expressions: happy, neutral, angry) × 2(feedback: gain, loss) repeated-measures ANOVA of the FRN average amplitudes showed a significant main effect of feedback, *F* (1, 22) = 11.135, *p* = 0.003, $\eta_{p}^{2}$ = 0.336, suggesting that loss (0.366 ± 0.427 µV) elicited a larger FRN amplitude than gain (1.246 ± 0.469 µV; *p* = 0.003). Both the main effect of expressions, *F* (2, 44) = 0.651, *p* = 0.526, and the interaction between expressions and feedback were not significant, *F* (2, 44) = 0.29, *p* = 0.796.

P3. A 3(expressions: happy, neutral, angry) × 2(feedback: gain, loss) repeated-measures ANOVA of the P3 average amplitudes showed a significant main effect of expressions, *F* (2, 44) = 4.019, *p* = 0.025, $\eta_{p}^{2}$ = 0.154, suggesting that angry faces (2.645 ± 0.544 µV) elicited a larger P3 amplitude than neutral faces (1.760 ± 0.513 µV; *p* = 0.007). However, happy faces (1.890 ± 0.525 µV) did not differ from angry or neutral faces (*p* > 0.05). Both the main effect of feedback, *F* (1, 22) = 0.015, *p* = 0.905, and the interaction between expressions and feedback were not significant, *F* (2, 44) = 1.712, *p* = 0.192 (see Figure 3).

## 4. Discussion

In the present study, we used ERPs to investigate the influence on the making-choices and feedback stages by facial expressions and gender. Behavioral results revealed that, for female trustees, happy or neutral faces elicited higher investment rates than angry faces (female: happy = neutral > angry), while, for male trustees, happy faces elicited the highest investment rates, followed by neutral faces, then angry faces (male: happy > neutral > angry). Electrophysiologically, during the making-choices stage, for female trustees, neutral faces elicited a larger N2 amplitude than angry faces (female: neutral > angry), while happy faces did not differ from angry or neutral faces. However, no such clear amplitude differences depending on the valence of facial expressions were seen for male trustees. Moreover, for distrust choices, happy faces elicited a larger P3 amplitude than neutral faces (distrust: happy > neutral), while angry faces did not differ from happy or neutral faces. However, no such clear amplitude differences depending on the valence of facial expressions were seen for trust choices (see Table 3). During the feedback stage, loss elicited a larger FRN amplitude than gain. Meanwhile, angry faces elicited a larger P3 amplitude than neutral faces, while happy faces did not differ from angry or neutral faces. Then, the meaning of both behavioral and EPR results were elaborately explained.

Regarding behavioral data, for female trustees, happy or neutral faces elicited higher investment rates than angry faces (female: happy = neutral > angry), while, for male trustees, happy faces elicited the highest investment rate, followed by neutral faces, then angry faces (male: happy > neutral > angry). The interaction between expressions and gender may stem from the fact that the subjective valence of neutral expressions is sensitive to contextual influences [33]. Previous studies found that neutral expressions were rated more positively in a positive context, while they were rated more negatively in a negative context [33]. In the present study, neutral expressions were rated more positively in female faces, while they were rated more negatively in male faces, as female and male faces themself represent positive and negative information, respectively [34]. Thus, the differential investment rates between happy faces and neutral faces were absent in female trustees; however, the differential investment rates between happy faces and neutral faces were observed in male trustees. This result was in line with a study showing that neutral female faces activated more positive networks and neutral male faces activated more negative networks [35].

On the electrophysiological level, during the making-choices stage, neutral faces evoked a larger N2 amplitude than angry faces when they appeared in female trustees (female: neutral > angry), while the N2 amplitude elicited by happy faces did not differ from that of angry or neutral faces. However, no such clear amplitude differences depending on the valence of facial expressions were seen for male trustees (male: happy = neutral = angry). Additionally, although the differential N2 amplitude was not significant, female trustees with neutral faces elicited a larger N2 than happy faces. The emotional content in female trustees, compared with male trustees, was elaborately discriminated. From this result, participants may tend to expect more positive feedback from female trustees compared with male trustees, as previous studies found that female trustees attracted more positive trustworthiness judgments [36,37]. Therefore, participants paid more attention to facial expressions of female trustees. However, it is difficult for participants to decide whether or not to cooperate with neutral faces of female trustees, as angry and happy faces are respectively associated with approach and avoidance behavior [6]. Therefore, when deciding whether to trust female trustees with neutral faces, participants were in a high-conflict situation, thereby leading to a larger N2, as the enhanced N2 is often related to cognitive conflict [38,39]. In contrast to female trustees, participants are less likely to expect positive feedback from male trustees [35,37], and this low level of expectation might impede our observation of the effect of facial expressions on male trustees.

During the making-choices stage, happy faces elicited a larger P3 amplitude than neutral faces for distrust choices, while angry faces did not differ from happy or neutral faces. However, for trust choices, no differences were found among angry, happy, and neutral faces in average amplitude. This probably occurred because participants were more likely to discriminate between the emotional content when they make a distrust choice relative to trust choice. One possible reason is that participants were less likely to consider other clues when they make trust choices, as trust choice is a default advantage choice [40,41]. In contrast, participants tended to process other clues when they made a distrust choice. Additionally, previous studies found that others’ happy expressions probably represent the intention of cooperation [6], and the enhanced P3 is related with prosocial motivation [20]. Therefore, happy faces elicited a larger P3 compared with neutral faces, when participants made a distrust choice.

During the feedback stage, loss induced a larger FRN amplitude than gain, which was in line with previous studies showing that negative outcomes elicited a larger FRN amplitude than positive outcomes [42,43]. However, the effect of facial expressions on the FRN amplitudes was absent. This result may suggest that facial expressions had no effect on the early component. Moreover, for the P3 component, the modulation of feedback was not observed during the feedback stage. The result was not consistent with some previous research showing that gain outcomes elicited a larger P3 amplitude than loss outcomes [24,25]. However, the result was in line with other research showing that the P3 encodes only the magnitude of reward feedback, not the valence of feedback [26,27]. In the present study, an answer was provided to solve the controversy for the functional significance of the P3 component. Critically, angry faces elicited a larger P3 amplitude than neutral faces, and the results showed that the modulation effect of facial expressions was observed on P3. Previous studies found that people were more sensitive to negative emotional stimuli than positive stimuli [44,45], hence the enhanced P3 elicited by angry faces compared with neutral faces, as it is generally thought to be related to motivational significance [46,47].

There are several limitations to this study that should be addressed. Firstly, participants always choose to invest all the money or not; therefore, it is impossible to investigate the potential interactions among investment amount, expressions, and gender. Secondly, gender was not investigated during the feedback stage due to insufficient trials, which should be confirmed in future studies. Despite both limitations, the present study offers novel insight into how others’ expressions and gender may influence the making-choices stage of the Trust Game. Thirdly, the sample size is rather small for making some bold conclusions. Therefore, it is necessary to expand the sample size in future research. Finally, participants should be recruited from more diverse backgrounds and different age groups in future research.

## 5. Conclusions

In conclusion, our findings provide new behavioral and electrophysiological evidence for the influence of others’ facial expressions on the making-choices and feedback stages. Most importantly, the effect of facial expressions on making choices and subsequent outcome evaluation was proved for the first time. Specifically, on the behavioral level, female trustees with happy or neutral faces elicited higher investment rates than angry faces. However, male trustees with happy faces elicited the highest investment rates, followed by neutral faces, then angry faces. Meanwhile, on the electrophysiological level, female trustees with neutral faces elicited more cognitive conflict (N2) than angry faces; no such pattern was observed when trustees were female. Additionally, during the making choices stage, happy faces elicited a larger P3 amplitude compared with neutral faces when participants made distrust choices; no such pattern was observed when participants made trust choices. During the feedback stage, we observed that loss elicited a larger FRN, and the P3 amplitude was modulated by facial expressions. The present study extended previous studies by instantiating both the making-choices and feedback stages.

## Figures and Tables

**Figure 1 ijerph-20-00568-f001:**
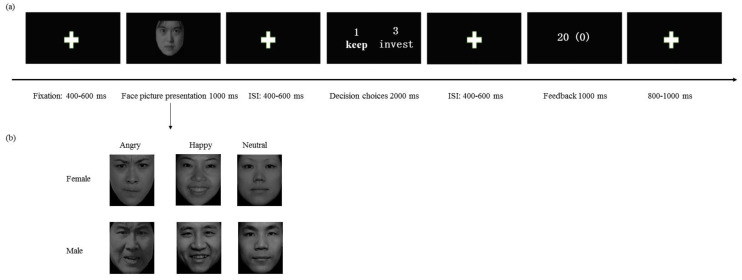
(**a**): Time sequence of stimuli in each trial of the decision task. Positions of 1 (distrust) and 3 (trust) for decision choices were counterbalanced between participants. Loss feedback was 0, and 20 was gain feedback. ISI = interstimulus interval; ITI = intertrial interval. (**b**): The photos of trustees differed regarding the emotional type (i.e., happy, angry, or neutral) and gender (i.e., female or male).

**Figure 2 ijerph-20-00568-f002:**
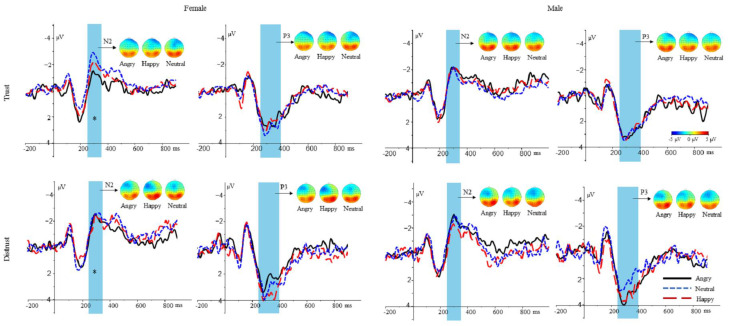
Grand averages according to the making-choices stages. The first row contains grand averages (N2 on electrode Fz and P3 on electrode Pz) for trusting decision, separated by happy, angry, and neutral faces of female or male trustees. The second row contains grand averages (N2 on electrode Fz and P3 on electrode Pz) for distrusting decision, separated by happy, angry, and neutral faces of female or male trustees. Vertical lines indicate the time windows that were quantified for statistical analyses. The topographic maps are also based on these time windows. *: *p* < 0.05.

**Figure 3 ijerph-20-00568-f003:**
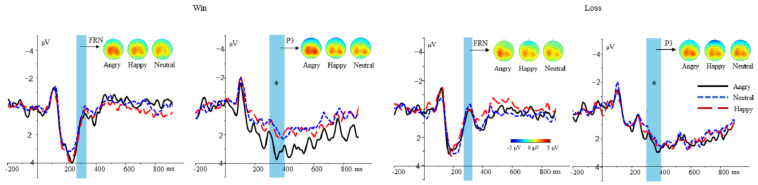
Grand averages according to the feedback stages. Grand average ERPs (FRN on electrode Fz and P3 on electrode Pz) for win and loss, separated by happy, angry, and neutral faces. *: *p* < 0.05.

**Table 1 ijerph-20-00568-t001:** Mean investment rates of different conditions.

		Gender	
		Female (M ± SE)	Male (M ± SE)
Expressions	Angry	0.352 ± 0.046	0.341 ± 0.048 **
	Happy	0.628 ± 0.038 **	0.673 ± 0.033 **
	Neutral	0.558 ± 0.037 **	0.545 ± 0.041 **

Female: happy > neutral, angry > neutral; Male: happy > neutral > angry. **: *p* < 0.01.

**Table 2 ijerph-20-00568-t002:** Mean reaction time (RT) of different conditions.

		Gender (ms)			
		Female (M ± SE)		Male (M ± SE)	
		Trust	Distrust	Trust	Distrust
Expressions	Angry	625.661 ± 46.992	609.089 ± 44.670	577.070 ± 47.340	580.896 ± 39.222
	Happy	607.483 ± 43.370	590.878 ± 50.637	591.09 ± 43.4427	605.768 ± 47.388
	Neutral	590.425 ± 48.634	587.797 ± 44.155	588.076 ± 41.269	592.391 ± 39.222

All *p* > 0.05.

**Table 3 ijerph-20-00568-t003:** The average mean amplitudes of N2 and P3 in the making-choices stage.

	Female	Male	Trust	Distrust
N2 (uv)				
Angry	−1.378 ± 0.436	−2.052 ± 0.492	−1.370 ± 0.456	−2.060 ± 0.548
Happy	−1.746 ± 0.482	−1.676 ± 0.567	−1.714 ± 0.408	−1.708 ± 0.663
Neutral	−2.216 ± 0.359 *	−1.912 ± 0.532	−1.915 ± 0.412	−2.214 ± 0.487
P3 (uv)				
Angry	1.963 ± 0.249	2.262 ± 0.329	2.122 ± 0.326	2.103 ± 0.311
Happy	2.228 ± 0.323	2.207 ± 0.323	2.046 ± 0.315	2.390 ± 0.337
Neutral	2.183 ± 0.308	1.920 ± 0.278	2.289 ± 0.311	1.814 ± 0.280

N2: Female: neutral > angry (* *p* < 0.05); P3: Distrust: happy > neutral (*p* = 0.07).

## Data Availability

The data that support the findings of this study are available upon request from the corresponding author, E.Z.

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
