# Peer review of "Others’ Facial Expressions Influence Individuals Making Choices and Processing Feedback: The Event-Related Potential and Behavioral Evidence"

_ijerph, 2022, doi:10.3390/ijerph20010568_

Round 1
Reviewer 1 Report
The authors have carried out an study that is currently having more attention towards understanding the facial expressions. However they are few concerns need to be addressed by the authors.
1) A comparison of the latest recent studies with the current study for concluding the novelty of the work. It is suggested to consider following reputed article from the nature journal "An ERP study on facial emotion processing in young people with subjective memory complaints".
2) In discussion section (Section 4), the authors discussed analysis clearly but the authors need to include tabular form for better readability and understanding of the analysis.
3)The novelty of the study need to be highlighted in abstract and conclusion section with future work.
4) The contributions and organization of the study must be included at the end of introduction section.
5) Kindly use the abbreviations in correct manner. In the abstract, the authors have not elaborated full form of ERP at the first instance. The same need to be verified in the rest of manuscript.
Reviewer 2 Report
Dear Editor after reviewing the manuscript, I have some minor comments that would improve the overall presentation of the work:
Please avoid abbreviations in the title, it is a general rule. Also, the first time you mention any abbreviation. no matter how familiar and generally accepted it is, you should introduce it properly.
Line 78, ’’…maximum amplitude of 250-320 ms.’’ I think this is a mistake.
Line 103-104, which local ethics committee, please provide the full name of the institution and decision number.
Line 143: I do not understand this ’’N=400’’ it is redundant I think.
Although the authors calculated the sample size I do think it is rather small for making some bold conclusions, maybe to tone down this through the text, also in the Study limitations section you should highlight at least the age and the background of subjects that are estimating the facial expression as additional factors. It is not the same if an elderly person is estimating this, vs a young adolescent one.
